# Study-Related Work and Commuting Accidents among Students at the University of Mainz from 12/2012 to 12/2018: Identification of Potential Risk Groups and Implications for Prevention

**DOI:** 10.3390/ijerph17103676

**Published:** 2020-05-23

**Authors:** Pavel Dietz, Jennifer L. Reichel, Antonia M. Werner, Stephan Letzel

**Affiliations:** 1Institute of Occupational, Social and Environmental Medicine, University Medical Centre of the University of Mainz, 55131 Mainz, Germany; jreichel@uni-mainz.de (J.L.R.); letzel@uni-mainz.de (S.L.); 2Department of Psychosomatic Medicine and Psychotherapy, University Medical Centre of the University of Mainz, 55131 Mainz, Germany; Antonia.Werner@unimedizin-mainz.de

**Keywords:** student, college, campus, accident, injury, epidemiology, student health

## Abstract

Background: Universities represent an important setting of everyday life for health promotion. The aim of the present study was to assess whether university students of specific disciplines might have an increased risk for having a study-related work accident and to analyze what types of study-related work accidents occur most frequently. Furthermore, knowledge regarding study-related commuting accidents will be provided by identifying places where study-related commuting accidents might occur most frequently and on potential types of commuting (walking vs. biking) which might be associated with an increased risk for having a study-related commuting accident. Methods: Retrospective analyses of a dataset provided by the Accident Insurance Fund of Rhineland-Palatinate, Germany, including all accidents that happened at the University of Mainz (JGU) between December 2012 and December 2018 were performed. Binominal tests were computed to reveal whether the frequency of study-related work accidents in students affiliated with a specific faculty or institution differs significantly from the expected frequency of all reported study-related work accidents. Results: Overall, 1285 study-related accidents were analyzed—of which, 71.8% were work and 28.2% commuting accidents. Students of ‘Faculty—Medicine’ (80.5%; *p* = 0.003), ‘Faculty—Chemistry, Pharmaceutical Sciences, Geography and Geosciences’ (90.7%; *p* < 0.001), and students that participated in study-related sports activities (97.4%; *p* ≤ 0.001) had a significantly increased risk for the occurrence of a study-related work accident. Needlestick and sharps injuries (NSIs) as well as lab accidents play a pivotal role. Furthermore, above 40% of the study-related commuting accidents were cycling accidents. Conclusions: There is a call for prevention in order to decrease the number of NSIs among medical students, lab accidents as well as sport-related accidents. Concrete implications for prevention are discussed in the present paper. In addition, given that students are among the most likely to bicycle, and given that most bicycle-related accidents involve fatal injuries, cycling safety campaigns need to be initiated on campus.

## 1. Introduction

According to the *Ottawa Charter for Health Promotion* by the World Health Organization (WHO), health is not just a state, but also ‘a resource for everyday life’. It is created and lived by people within the settings of their everyday life—where they learn, work, play, and love [1] —emphasizing the interconnectedness between individuals and their environments. In 2015, an international expert group formulated the *Okanagan Charter* and identified universities as being an important setting of everyday life for health promotion. They further stated that from a public health point of view, the collective of university students would be of particular relevance. They argued that health promotion in university students would not only be beneficial to the health of the target population (students), but since university students are the executives, decision makers and also parents of tomorrow, health promotion in this population may also benefit the general society [2]. Again in 2015, the German Government passed the so called prevention law (Präventionsgesetz), aiming to strengthen health promotion and prevention in the different settings of everyday life [3]. According to this law, the statutory health insurances have to spend a set amount of money for each insurant for health promotion and prevention projects in settings of everyday life. Supported with financial resources of the prevention law, the Healthy Campus Mainz project was initiated in 2018. It is an interdisciplinary research project aiming to create, implement, and evaluate an evidence-based, sustained, and holistic health management program for the approximately 32,000 students at the Johannes Gutenberg University of Mainz (JGU).

Due to their considerable consequences on health, accidents are one of the most important public health problems [4]. In Germany, approximately 10 million accidents happen each year—of which, approximately 24,500 result in death [5]. This overall estimation for Germany is based on a summary of five individual accident statistics including accidents that occur (i) during leisure time (3.9 million), (ii) at home (3.2 million), (iii) at school (1.3 million), (iv) at work (1 million), and (v) in traffic (0.4 million). Although approximately 2.9 million students (winter term 2018/2019) are registered at German universities [6], there is only limited knowledge about study-related accidents among university students in Germany, defined as accidents that occur during any activity in the context of studying (e.g., at seminars, in the lab, and during study-related sports activities), called study-related work accidents, as well as commuting accidents that occur on the way between two courses or seminars or on the way to or from university [7,8]. The very small number of studies from other European countries dealing with this topic typically describe accident rates among university students assessed by student surveys. However, the studies show a substantial variation in these rates. For example, a survey among 617 randomly selected 3rd year students from the University of Helsinki, Finland revealed a total accident rate of 28.7% (*n* = 177) for the last three years. Of these accidents, almost half happened during sports activities and 14% in traffic. Only 0.5% were classified as having occurred during activities directly related to studies or in the university environment [9]. Another survey among 1208 higher education students from the UK revealed that 18% (*n* = 222) of the students had at least one injury during the last year requiring medical attention, while 4% reported an injury which was related to their studies [10]. A more recent multicenter cross-sectional health survey conducted at 16 institutions of higher education in North Rhine-Westphalia, Germany, reported that 8.8% (252 out of 2855) of the participants had experienced a study-related accident. Of all these accidents, approximately 60% took place during study-related sports activities and nearly one-quarter on the way to or from university. Only a few accidents occurred in the university environment (e.g., on the floors, lecture rooms or on stairways) [7]. These examples demonstrate the variety of reported accident rates, which might be due to the heterogeneity of the survey methodologies used for the studies. Furthermore, questions regarding specific types and reasons for study-related accidents as well as the question around which particular disciplines might have an increased risk of having a study-related accident are rarely examined. Only very few studies have examined the frequency of specific types of study-related accidents among students of specific disciplines. For example, surveys performed among medical students have indicated that approximately one-quarter of medical students had a history of needlestick and sharps injuries (NSIs) [11,12,13], i.e., injuries to the skin by handling sharp instruments by which blood of patients may be transmitted. However, such studies with focus on a specific student collective do not allow any conclusion regarding potential student collectives of increased risk for the occurrence of a study-related accident. Regarding study-related commuting accidents, from a public health point of view, a special focus should be on cycling accidents. Because on the one hand, university and college students were stated to be among the most likely to bike [14] and commute to their campuses and universities daily in concentrated multimodal transportation systems comprised of other road users [15]. On the other hand, bicycle-related accidents very often involve fatal injuries such as head injuries [16].

In conclusion, although approximately 2.9 million students are enrolled at German universities and although accidents are one of the most important public health problems, there is still a large knowledge gap on the topic of accidents among university students, especially with regard to potential risk groups (e.g., students of specific disciplines) as well as reasons for study-related accidents. The little information that is available, in most cases, is based on student surveys using self-report survey techniques. In addition, to the best of our knowledge, we are not aware of any study analyzing independent and complete accident reports for university students from a university in Germany. However, from a public health point of view, a central issue is whether preventive activities are best aimed at groups held to be at high risk or at the population as a whole [17]. Therefore, the evidence-based identification of potential risk groups—in our case, students of specific disciplines that might have an increased risk for the occurrence of a specific study-related accident—plays a central role. As part of the Healthy Campus Mainz project, we address this issue. The provided knowledge may be helpful to guide decision makers and protagonists of health promotion at universities to address actions to prevent study-related accidents in a more individualized way. Therefore, using accident data provided by the Accident Insurance Fund of Rhineland-Palatinate (Unfallkasse Rheinland-Pfalz), the present study aimed to: (i)Assess whether students of specific disciplines might have an increased risk for having a study-related work accident and to identify what types of accidents these are.(ii)Provide knowledge on places where study-related commuting accidents might occur more frequently (e.g., on the street vs. on the stairs inside a building).(iii)Identify potential types of commuting (walking vs. biking) that might be associated with an increased risk for having a study-related commuting accident.

## 2. Materials and Methods 

### 2.1. Basic Population

The JGU is a full university and the biggest institution of tertiary education in Rhineland-Palatinate administrating ten faculties, the School of Music Mainz, the School of Art Mainz, and the International Preparatory and Language Center that exists since 2012 as a fusion of the formerly separate central institutions Language Center and Preparatory College. The faculties are the basic organizational units of the JGU fulfilling the tasks assigned to the university in the fields of research, learning, and continuing education for their respective areas. Each faculty represents specific academic disciplines and is chaired by its own dean [18]. Appendix A presents the number of registered students at the JGU distributed for the different faculties and schools as well as the proportion of female students for the winter terms 2012/2013 until 2018/2019 according to the annually published Data and Statistics Reports of the JGU [19]. It shows that from 2012/2013 (*N* = 36,440) to 2018/2019 (*N* = 31,967), the number of students is slightly but continuously decreasing, with a relatively constant proportion of female students of approximately 59%. At the JGU, all study-related sports activities are administered and organized by the local office of the General University Sport. It offers the students a variety of different sports activities ranging from ball and setback sports to gymnastics, dancing, climbing, yoga, and tai chi. These activities take place in supervised sport groups.

### 2.2. Dataset

The German Social Accident Insurance (Deutsche Gesetzliche Unfallversicherung) is one of five mandatory insurances within the German Social Security System. Employees, children and students are insured during their activities at the workplace, in schools, nursery schools and institutes of higher education. Other groups such as domestic staff and voluntary workers are also insured. It is the umbrella organization of 16 local Accident Insurance Funds [20]. Any study-related accident that occurs to students that results in treatment by a doctor or death has to be reported to the responsible local Accident Insurance Fund. For study-related accidents that happen to university students at the JGU, the Accident Insurance Fund of Rhineland-Palatinate (Unfallkasse Rheinland-Pfalz) is responsible. For the present paper, we retrospectively analyzed data of all reported accidents that happened at the JGU between December 2012 and December 2018 provided by the Accident Insurance Fund of Rhineland-Palatinate. To protect the privacy of the insured persons, the dataset did not include any personally identifiable information (e.g., name, birthday or date of the accident). The following variables were provided and analyzed for the present paper: (i) *gender* (f/m/ d), (ii) *age*, (iii), *institution (faculty/school/sport)*, (iv) *type of accident (work or commuting)*, (v) *type of study-related work accident*, (vi) *place of study-related commuting accident* (e.g., on the side walk or on the stairs inside a building), *type of study-related commuting accident* (e.g., bicycle or car). Since our group received the whole anonymous dataset from the Accident Insurance Fund of Rhineland-Palatinate to perform the analysis described in the present paper, a formal agreement from the local ethical committee was not required.

### 2.3. Statistical Analysis

The results described in the present paper are mostly descriptive. We present the overall number of accidents among students at the JGU between December 2012 and December 2018. We further specified the type of study-related accident (work or commuting accident) and within latter, report the more specific type of accident (such as injuries caused by falling, a bump, or a mechanical impact). Accident frequencies are presented as absolute numbers and relative percentages. To identify potential areas of risk, we defined the total number of reported accidents as reference population regarding the frequency of study-related work and commuting accidents. Binominal tests [21] for each faculty and University institution (such as General University Sports) were computed to reveal whether the frequency of study-related work accidents in students affiliated to a specific faculty or institution differs significantly from the expected frequency of all reported study-related work accidents. In this regard, the null hypothesis assumes that the two categories (work vs. commuting accident) should occur to the same extent as in the overall accident population. In case of a significant difference, students of the identified institution were more likely to have had a study-related work or commuting accident. As we wanted to particularly look into the frequency of study-related work accidents in order to detect potential areas for study-related work accident prevention at the university, only differences with regard to *more study-related work accidents* were interpreted following our study aim (ii). For the possibly identified areas of more frequent study-related work accidents, we will provide descriptive data for the specific type of study-related work accident and the reason for the accident. Regarding study-related commuting accidents, we calculated absolute and relative frequencies for the reported *place* where the commuting accident happened (e.g., on the street vs. on the stairs inside a building) and *type of commuting* (e.g., walking vs. biking).

## 3. Results

Overall, 1285 study-related accidents were reported among students at the JGU between December 2012 and December 2018. The mean age of the affected students was 23.9 years (*SD* = 4 years) and 56.2% of students (*n* = 722) were female. Almost ¾ of all reported study-related accidents were work accidents (*n* = 922; 71.8%) and 28.2% (*n* = 363) were commuting accidents (Table 1). In Appendix A
Appendix A, the types and reasons for study-related commuting and work accidents are described. In short, of all study-related work accidents, the most frequently reported accident was an “injury by fall” (*n* = 344; 37.3%), followed by an “injury by a bump or hit” (*n* = 206; 22.3%) as well as “injuries caused by mechanical impact” (*n* = 204; 22.1%). Injuries by physical, chemical, or biological impact occurred in 8.1% (*n* = 75). Regarding the study-related commuting accidents, the most frequently reported type of accident was an “injury caused by falling” (*n* = 199; 54.8%) as well, followed by “injuries by a bump or hit”, with 41.0% (*n* = 149). No accident resulted in death. Over the years, the overall accident rate (number of accidents per year/students per year) was 0.8%–0.6% were study-related work accidents and 0.2% were commuting accidents (Appendix A).

### 3.1. Specific Disciplines with an Increased Risk for Having a Study-Related Work Accident

Binominal tests revealed that students of ‘Faculty 04—Medicine’ (80.5%; *p* = 0.003), ‘Faculty 09—Chemistry, Pharmaceutical Sciences, Geography and Geosciences’ (90.7%; *p* < 0.001), and students that participate in study-related sports activities administered by the General University Sport (97.4%; *p* ≤ 0.001) had a significantly higher proportion of study-related work accidents (Table 2). 

Looking more closely at these three institutions (Table 3), we observed that of all reported study-related work accidents at Faculty 04 ‘pricking oneself on something/being pricked by someone’ was, at 66.5% (*n* = 107), the most frequently reported reason for a study-related work accident. At Faculty 09, the most frequently reported reason for a study-related work accident was an ‘injury by hazardous substance’ (*n* = 28; 41.2%). Further, approximately one-third of study-related work accidents at Faculty 09 were caused by ‘cutting oneself with or on something’ (*n* = 20; 29.4%). Finally, in General University Sport, the most frequently reported reason for an accident was ‘twisted one’s ankle’ (*n* = 120; 28.9%) followed by ‘got hit by something’ (*n* = 68; 16.4%).

### 3.2. Places and Types of Commuting Accidents 

Of all 363 reported commuting accidents among university students at the JGU, almost two-thirds occurred on the ‘street’ (*n* = 226; 62.3%) and one-quarter on the ‘side walk’ (*n* = 94; 25.9%). Only twelve accidents (3.3%) happened on the stairs inside a university building. Regarding the type of commuting which was used when a commuting accidents occurred, most accidents were ‘bike’ accidents (*n* = 147; 40.5%) followed by accidents while ‘walking’ and taking the ‘car’ (*n* = 98; 27% and *n* = 61; 16.8%, respectively). All places of study-related commuting accidents and the type of commuting are presented in Table 4. 

## 4. Discussion

The present study aimed to provide deeper insights into a relatively underrepresented research topic, namely study-related accidents among university students. More specifically, we assessed whether students of specific disciplines might have an increased risk for having a study-related work accident. Furthermore, we provided knowledge on study-related commuting accidents. In contrast to previous studies exploring this topic [7,9,10], we did not use self-reported survey data. Instead, objective accident data provided by the Accident Insurance Fund of Rhineland-Palatinate were analyzed retrospectively.

### 4.1. Study-Related Work Accidents

The overall number of reported study-related work accidents was 3-fold higher than the number of commuting accidents. Most of these study-related work accidents took place during study-related sports activities. This supports the results of previous studies based on student surveys [7,9]. This relative high proportion of accidents during study-related sports activities might be due to the fact that at the JGU, approximately 10,000 students participate in study-related sports activities administered by the General University Sport of Mainz every week. Preferred sports activities are soccer, handball, basketball, volleyball, rugby, or field hockey (unpublished figures provided by the German University Sport Mainz office). According to an epidemiological study by Henke et al. (2014) analyzing 200,884 sports-related injuries between 1987 and 2012 in Germany, approximately two-thirds of all injuries were reported in soccer, handball, basketball, and volleyball. They concluded that ball sports are still a clear focal area for injury prevention, as participation and injury risk are highest in this group [22].

Furthermore, medical students were observed to have an increased risk for study-related work accidents. Almost three-quarters of all accidents that occurred among medical students were NSIs. A previous anonymous electronic survey among surgical personnel revealed that 22% of all surveyed medical students had a history of a NSI [12]. A comparable high prevalence for NSIs among medical students (28% and 33.7%, respectively) was observed within investigations by Bernard et al. (2013) and Ghasemzadeh et al. (2015) [11,13]. Within the latter mentioned survey, vein puncture was the most common mechanism of injury (24.3%), followed by drawing arterial blood (20.3%) and injections (7.4%). In this context, Siegmann et al. (2016) took a closer look at the time when NSIs occur during medical studies. While only 20.6% of the students indicated a NSI at the beginning of their studies, half of the students (50.9%) had experienced at least one injury at the end of the clinical period [23]. This finding is consistent with the results of previous studies indicating that students of higher semesters are more prone to injuries because of having more clinical activities, which increases the possibility of injury [24,25]. NSIs happened most frequently in surgical units, in internal medicine, and in gynecology [23]. Our study as well as the reported findings of other groups demonstrate that NSIs are a prevalent matter among medical students. Since NSIs may cause the transmission of many blood-borne infections and diseases such as hepatitis C, B, and HIV, which pose a substantial health risk to the practitioner and the patient [13], there is an important call for prevention in order to increase safety of medical students and patients. Ghasemzadeh et al. (2015) recommended that holding workshops and increasing medical students’ awareness and skills to face the risks of NSIs may be effective in mitigating them. In addition, the American College of Surgeons (ACS) recommended the use of double gloving, hands-free zone, and blunt-tip suture needles for decreasing NSIs [26]. In contrast to that, a study on preventing NSIs among medical students at the University Medical Center Düsseldorf, Germany, showed that both intensive safety trainings of medical students as well as the implementation of so-called safe instruments did not led to a reduction in NSIs [23]. In conclusion, more research is needed to address the environmental and individual factors that might predict the occurrence of NSIs among medical students in order to create and implement efficient, target-group (students of higher semesters and of specific disciplines) and setting-specific prevention programs.

The third group, which reported a higher proportion of study-related work accidents, were students of the JGU Faculty 09 including the disciplines Chemistry, Pharmaceutical Sciences, Geography and Geosciences. One specific characterization of the disciplines Chemistry, Pharmaceutical Sciences, and Geosciences is that parts of their studies take place in the lab. Consequently, almost one-half of the accidents that occurred among students at Faculty 09 were injuries by physical, chemical or biological impact (41.2% injured by hazardous substance; 7.4% burned/scald on something). The epidemiology of accidents in academic chemistry laboratories is not new (e.g., [27,28]), but with respect to university students, it is a rarely investigated topic. However, a recent study from the US evaluating the trends in laboratory-related injuries at IOWA State University from 2001 to 2014 revealed that students (a mixed sample of graduate assistants and student employees) represented the most frequently injured group, compromising above 40% of the laboratory-related injuries [29]. An analysis of the current status on laboratory safety management of different universities, research institutes and industrial research institutions revealed that more than three-quarters of laboratory accidents happened in the field of chemical research and operation management. Furthermore, they observed that more than half of the accidents took place due to careless use of dangerous chemicals and careless use of mechanic instruments. Most accidents were caused by negligence of researchers [30]. Therefore, we follow the argumentation by Walters et al. (2017) recommending that the implementation of training on chemical safety in the academic laboratory environment is an important step for reducing the incidence of lab-related accidents and for preparing students for the working environment [31]. In addition to a human behavior approach focusing on safety education, the implementation of an institutional lab safety management is recommended in the literature [30], which also addresses environmental conditions for reducing lab-related accidents. However, such an undertaking, especially at a university, depends on the availability of human resources and budget.

### 4.2. Study-Related Commuting Accidents

Focusing on study-related commuting accidents, the vast majority of accidents happened outside. Only twelve accidents happened on the stairs inside a building which stands in contradiction to the quite high number of reported injuries and deaths by stair-related accidents in the general population [32,33]. The relatively low number of only twelve reported stair-related accidents in 6 years might be due to two reasons. First, we addressed a relatively homogenous collective according to age (young to middle aged) and education, namely university students, which might have a potentially lower risk for the occurrence of a stair-related accident. Secondly, there might be a potential bias of reporting due to a lack of knowledge regarding the responsibility when a study-related commuting accident occurs. Therefore, the reported number of 12 stair-related accidents and the total number of reported accidents described in the present paper might be underestimated.

Above 40% of the study-related commuting accidents were cycling accidents. From a public health point of view, this number puts the commonly recommended use of active transportation into a different perspective. Of course, as has been demonstrated in numerous studies, physical activity, which is defined as any bodily movement produced by skeletal muscles that requires energy expenditure [34], is associated with plenty of positive effects on physical and psychological health. For example, it reduces the risk of cardiovascular, metabolic, and mental diseases, lowers the risk of many forms of cancer, improves fitness, supports weight management, and increases an individual’s chance of living longer [35,36]. However, as Jaffe (2019) wrote in the Lancet: as more bicycle riding is encouraged for health and environmental (fighting climate change and traffic) reasons, an increase in cycling accidents and deaths is causing a public health dilemma. Given that university and college students are among the most likely to bike [14] and commute to their campuses and universities daily in concentrated multimodal transportation systems comprised of other road users [15], and given that most bicycle-related accidents involve fatal injuries such as head injuries [16], cycling safety campaigns need to be initiated on campus [37]. According to a review of bicycle safety campaigns from the US, emotional campaigns that depend on fear would often be more effective at increasing safety than informational laws, suggested behavior, etc., campaigns. However, they further stated that bicycling already has a strong association with fear in the US, which discourages more people from riding bikes. Aiming to avoid fear-based emotional campaigns, they concluded that safety campaigns that personalize and humanize cyclists would be ideal [38]. Furthermore, bicycle safety campaigns may address the following issues including infrastructural and behavioral aspects: (i) ensure that the bicycle is in working order (e.g., breaks), (ii) enhance the visibility of cyclists to other road users, (iii) enhance the ability of cycling safe (e.g., trainings and traffic rules), (iv) sensitize cyclists to wear helmets, and (v) not to drink alcohol when cycling [39,40,41]. Furthermore, within their critical review on the safety impacts of bicycle infrastructure including literature on 22 bicycle treatments, DiGioia et al. (2017) formulated some defensible conclusions regarding the safety and effectiveness of certain bicycle treatments, such as bike lanes and removal of on-street parking. They further stated that the vast majority of bicycle safety studies vary greatly in sample sizes, controls, and statistical rigor and recommended that further research needs to be conducted investigating safety impacts of bicycle infrastructure [42]. 

### 4.3. Limitations

A primary limitation of the present paper is the potential lack of reporting due to the fact that students may not know that study-related accidents are classified as ‘study related’ and have to be documented by a doctor and reported to the responsible local Accident Insurance Funds (also by the doctor). This may have led to an underestimation of the true number of study-related accidents among students, especially with regard to commuting accidents. Consequently, the accident rates presented in the present paper have to be interpreted with caution. Furthermore, the dataset used for the present paper was limited to a small number of accident-specific data. Additional interesting information such as potential predictors of accidents and sociodemographic variables were not collected.

## 5. Conclusions

In conclusion, the present study aimed to address a relatively rarely investigated topic, namely study-related work and commuting accidents among university students. Using complete and objective accident data provided by the Accident Insurance Fund of Rhineland-Palatinate, three potential risk groups for study-related work accidents were identified: sporting accidents among students participating in study-related sport activities, NSIs among medical students as well as laboratory accidents among students of the natural sciences. From a public health point of view, this evidence-based identification of potential risk groups plays a central role to guide decision makers and protagonists of health promotion at universities to address actions to prevent study-related accidents in a more individualized way [17]. In order to create awareness for this relatively ‘disregarded’ topic and to be able to transfer the present results into preventive actions at the JGU, the next steps of the Healthy Campus Mainz project will be to communicate these findings to the university management, to the deans of the different faculties as well to the protagonists of health promotion at campus. With regard to the high percentage of cycling accidents, further research needs to be conducted investigating the environmental and personal conditions of the campus Mainz and the students at the JGU that may cause cycling accidents.

## Figures and Tables

**Table 1 ijerph-17-03676-t001:** Accident figures and sample characteristics.

Total Number of Accidents, *N*	1285
Type of accident, *N* (percentage)	
Commuting	363 (28.2)
Work	922 (71.8)
Gender, *N* (percentage)	
Female	722 (56.2)
Male	563 (43.8)
Age, range (mean ± SD)	17–54 (23.9 ± 4.0)

**Table 2 ijerph-17-03676-t002:** Total number of study-related accidents, commuting accidents and work accidents distributed for the different institutions at the JGU (*N* = 1285).

Faculty/School at JGU	Total Number of Accidents *N* (%)	Type of Accident *N* (%)	
		Commuting	Work
F 01—Catholic and Evangelic Theology	1 (0.1)	1 (100)	0
F 02—Social Sciences, Media and Sport	18 (1.4)	10 (55.6)	8 (44.4)
F 03—Law and Economic Sciences	7 (0.5)	3 (42.9)	4 (57.1)
F 04—Medicine	200 (15.6)	39 (19.5)	161 (80.5)
F 05—Philosophy and Philology	12 (0.9)	8 (66.7)	4 (33.3)
F 06—Translation, Linguistic and Cultural Sciences	26 (2.0)	11 (42.3)	15 (57.7)
F 07—History and Culture Sciences	3 (0.2)	3 (100)	0
F 08—Physics, Mathematics and Computer Sciences	12 (0.9)	6 (50)	6 (50)
F 09—Chemistry, Pharmaceutical Sciences, Geography and Geosciences	75 (5.8)	7 (9.3)	68 (90.7)
F 10—Biology	15 (1.2)	6 (40.0)	9 (60.0)
School of Music	6 (0.5)	2 (33.3)	4 (66.7)
School of Art	2 (0.2)	0	2 (100)
General University Sport	426 (33.2)	11 (2.6)	415 (97.4)
Faculty unknown	482 (37.5)	256 (53.1)	226 (46.9)
Total	1285 (100)	363 (28.2 *)	922 (71.8 *)

F, faculty; JGU, Johannes Gutenberg University; * percentage of total number of accidents (*N* = 1285).

**Table 3 ijerph-17-03676-t003:** Types of and reasons for study-related work accidents distributed for the institutions with a significantly increased risk for the occurrence of a study-related work accidents.

Faculty/School	Type of Accident	Reason for Accident	*N* (%)
F 04—Medicine (*N* = 161)	Injured by mechanical impact	Pricked oneself on something/pricked by someone	107 (66.5)
		Cut oneself with/on something	10 (6.2)
		Other	7 (4.4)
	Injured by bump/hit	Hit by something	6 (3.7)
		Banged on something	5 (3.1)
		Other	2 (1.2)
	Injured by physical, chemical or biological impact	Harmed by virus/pathogen/carrier	5 (3.1)
		Burned/scalded oneself on something	2 (1.2)
		Injured by hazardous substance	2 (1.2)
	Other	Diverse	15 (9.2)
F 09—Chemistry, Pharmaceutical Sciences, Geography and Geosciences (*N* = 68)	Injured by physical, chemical or biological impact	Injured by hazardous substance	28 (41.2)
		Burned/scalded oneself on something	5 (7.4)
	Injured by mechanical impact	Cut oneself with/on something	20 (29.4)
		Other	3 (4.4)
	Injured by fall	Twisted one’s ankle	3 (4.4)
		Other	3 (4.4)
	Other	Diverse	6 (8.8)
General University Sport (*N* = 415)	Injured by fall	Twisted one’s ankle	120 (28.9)
		Miscarried rotation	49 (11.8)
		Fallen/tripped over something	40 (9.6)
		Other	21 (5.1)
	Injured by bump/hit	Hit by something	68 (16.4)
		Collided with someone/something	40 (9.6)
		Banged on something	23 (5.5)
		Other	5 (1.2)
	Other	Diverse	49 (11.8)

F, faculty; JGU, Johannes Gutenberg University.

**Table 4 ijerph-17-03676-t004:** Cross-tabulation for the variables ‘place where the study-related commuting accidents happened’ and ‘type of commuting’ (*N* = 363).

	Place of Accident, *n*	On the Street (*n* = 226; 62.3%)	On the Side Walk (*n* = 94; 25.9%)	On the Stairs Inside a Building (*n* = 12; 3.3%)	At the Station/Bus Stop (*n* = 11; 3.0%)	Other (14)/Not Reported (6) (*n* = 20; 5.5%)
Type of Commuting, *n*						
Bike (*n* = 147; 40.5%)		133	14	-	-	-
Walking (*n* = 98; 27.0%)		10	75	12	-	1
Car (*n* = 61; 16.8%)		61	-	-	-	-
Public transport (*n* = 14; 3.9%)		14	-	-	-	-
Motorcycle (*n* = 7; 1.9%)		7	-	-	-	-
Skateboard (*n* = 4; 1.1%)		-	4	-	-	-
Other (1)/not reported (31) (*n* = 32; 8.8%)		1	1	-	11	19

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
