# Peer review of "Study-Related Work and Commuting Accidents among Students at the University of Mainz from 12/2012 to 12/2018: Identification of Potential Risk Groups and Implications for Prevention"

_ijerph, 2020, doi:10.3390/ijerph17103676_

Round 1

Reviewer 1 Report

The paper presents the analyses of a data base of accidents that university students at the University of Mainz experienced from 12/2012 to 12/2018. The paper is well written and easy to read. Unfortunately there is just very little "there" there. 

The motivation and reasoning for the study is not well established at all. What evidence is there that knowing the "study-related" differences in accident rates will help prevent accidents?

What does "study-related" mean and why does it matter? The paper goes back and forth from referring to the students' "work" as "work" and "study-related. There is one point, later in the paper where the authors define "study-related" but they provide no guidance as to why incident rates might be different for these categories and more importantly, why that would matter. 

This is particularly disappointing given the VERY important findings (that the authors reference later) about laboratory (both chemical and medical) and sports related safety issues on campus. This could have motivated the paper somewhat or somehow but is not even considered. 

As an American, I did struggle with the use of the term "faculty" as I am most familiar with that term referring to an institution. In this paper, I could not tell if it referred to a "facility" or, what I would call an academic department. A quick clarification on this would be helpful. 

Finally, your finding about cycling is VERY interesting and probably the most noteworthy of the paper. I would love to see the paper reorganized to have the introduction actually talk the risks of the different types of accidents in the general population and how these may or may not be different at universities and what this might mean for university administration and safety policies. Then you could compare your findings to this.

A paper with a structure similar to this would have some motivation to it. Right now, it reads as an attempt to get a publication after you needed to do some analysis on a big data set.  

Author Response

Responses to Reviewer 1

Comment 1: The paper presents the analyses of a data base of accidents that university students at the University of Mainz experienced from 12/2012 to 12/2018. The paper is well written and easy to read. Unfortunately there is just very little "there" there 

Response: Thank you very much for your positive words and constructive comments on our manuscript. We are grateful for the detailed suggestions that were very helpful for improving it. We hope that you will be satisfied with the revised version in which we have incorporated your points. If you should have any further recommendations for improving our manuscript, please communicate these to us. Please notice that we included additional references into the manuscript. Therefore, the numbers and order of the references in the text as well as in the reference list changed. These changes are not marked in the manuscript in order to improve readability for the reviewers. For the purpose of your traceability, whenever a reference has been added to the manuscript, it is mentioned in the respective response to your comment. 

Comment 2: The motivation and reasoning for the study is not well established at all. What evidence is there that knowing the "study-related" differences in accident rates will help prevent accidents? 

Response: From a public health point of view, a central issue is whether preventive activities are best aimed at groups held to be at high risk or at the population as a whole. Such a decision is often an issue in discussions about policies relating to vaccinations, health promotion, infectious disease control as well as protection from risks relating to toxic agents or radiation, etc. For example, during the current debate around the COVID-19 pandemic, it has been communicated that people of higher age may be at increased risk for having a severe course of disease. Thus, older people have been characterized as a potential risk group. The same has been stated for people with pre-existing conditions such as cardiovascular or lung diseases. Within the present paper, we aimed to identify student collectives that may be at increased risk for the occurrence of a study-related accident, in order to address activities to prevent specific accidents (for example NSIs) to the right collective (for example medical students).However, we totally agree with you that this point becomes not clear by reading the introduction. Therefore, we now added a new paragraph to the manuscript (line 106 – 113) addressing this issue. Furthermore, we deleted the last sentence of the introduction. 

Comment 3: What does "study-related" mean and why does it matter? The paper goes back and forth from referring to the students' "work" as "work" and "study-related. There is one point, later in the paper where the authors define "study-related" but they provide no guidance as to why incident rates might be different for these categories and more importantly, why that would matter.

Response: Thank you very much for that comment. We totally agree with you that we have to keep the wording and terminology consistent through the whole manuscript and that the term “study-related accidents” has to be defined earlier in the manuscript. In this context, another term which has been used in the literature to describe “study-related accidents” is “accidents in context of study” (see the paper by Faller et al., 2010: https://www.ncbi.nlm.nih.gov/pubmed/20159071). However, our native-speaker who checked the manuscript recommended to use the more common term “study-related” in order to improve readability.

For clarification, the term “study-related work accident” describes all accidents that occur during any study-related activity/study-related “work” (e.g. in the lab or in the lecture room). Following your suggestion, we now added a specified definition and corresponding references into the introduction of the manuscript (line 66- 69). Furthermore, we deleted the “old” description from the methods section (line 145 – 148). In addition, to keep the wording consistent through the whole manuscript, we now consistently use the terms “study-related accident”, “study-related work accident”, and “study-related commuting-accident”. Please see our changes through the whole manuscript.

Comment 4: This is particularly disappointing given the VERY important findings (that the authors reference later) about laboratory (both chemical and medical) and sports related safety issues on campus. This could have motivated the paper somewhat or somehow but is not even considered.

Response: During manuscript preparation, we considered not to anticipate the literature reported in the discussion section because what all of these studies have in common, in contrast to our study, is that they focus on a specific student population, for example, on medical students. Therefore, these studies do not allow to draw a conclusion on potential student population with an increased risk for the occurrence of a study-related work accident (e.g. medical students have an increased risk for the occurrence of a study-related work accident compared to students of the humanities). Nevertheless, the results of these studies are suitable to compare the results of our study with regard to a potential identified specific risk group. Consequently, we decided not to mention these papers in the introduction but in the discussion section of the manuscript.

However, with regard to your comment, we now added the topic NSIs among medical students as an example into the introduction in order to make our point clear for the reader (line 87 - 93). As a consequence, the definition and abbreviation of NSIs is now introduced earlier in the manuscript and has been deleted in the discussion section.

Comment 5: As an American, I did struggle with the use of the term "faculty" as I am most familiar with that term referring to an institution. In this paper, I could not tell if it referred to a "facility" or, what I would call an academic department. A quick clarification on this would be helpful.

Response: Thank you very much for the hint that Americans may struggle with the term faculty. For clarification, the ten faculties of the Johannes Gutenberg University of Mainz (JGU) are the basic organizational units of the JGU fulfilling the tasks assigned to the university in the fields of research, learning, and continuing education for their respective areas. Each faculty represents specific academic disciplines and is chaired by its own dean. To make this point clear for the reader, we now included this information into the manuscript and added an additional reference (line 128 - 131), where the organization of the JGU as well as the role of the faculties are described in detail. The reference is in English language: https://faculties.uni-mainz.de/

Comment 6: Finally, your finding about cycling is VERY interesting and probably the most noteworthy of the paper. I would love to see the paper reorganized to have the introduction actually talk the risks of the different types of accidents in the general population and how these may or may not be different at universities and what this might mean for university administration and safety policies. Then you could compare your findings to this.

Response: Thank you very much for this comment. And yes, we agree that the comparison of accident data between university students and the general population is a very interesting question. However, our a priori defined research question was to assess whether students of specific disciplines might have an increased risk for having a study-related accident and to identify what types of accidents these are. Based on our student-focused perspective as researchers involved in the Healthy Campus Mainz project, we structured the introduction of the manuscript as follows: i) description of the setting of interest (university students), ii) description of the public health relevance of accidents, iii) description of the lack of knowledge with regard to potential risk groups, iv) description why this is relevant (see your comment number 2), and v) description of the aim of the study. Therefore, we sincerely hope that you accept our decision not to change the whole structure of the introduction.

However, following your recommendation on introducing the topic of cycling accidents already in the introduction of the manuscript, we now added a new paragraph highlighting the public health relevance of this issue into the introduction (line 93 – 98).

Comment 7: A paper with a structure similar to this would have some motivation to it. Right now, it reads as an attempt to get a publication after you needed to do some analysis on a big data set.

Response: Once again, we hope that we addressed all comments to your satisfaction and that the revised version of the manuscript, and especially the introduction, are now suitable for publication. In particular, we hope that you accept our argumentation with respect to your comment number 6. Should you have any further comments for possible improvements, please communicate these to us.

Reviewer 2 Report

The introduction should be further developed. It is too short to support the findings, and is still poorly explored with the international literature.

The discussion is also poorly explored with the international literature. This section should be improved in order to explore and discuss the main findings with other works reviewed in section 1 (1. Introduction).

Line 307: should be standardized throughout the text. [36–38] should be [36,38]

The conclusion must be deepened. It is very poor and should be further developed.

Authors should follow Instructions for Authors of the journal Int. J. Environ. Res. Public Health.

Author Response

Responses to Reviewer 2

Comment 1: The introduction should be further developed. It is too short to support the findings, and is still poorly explored with the international literature. 

Response: Thank you very much for your constructive comments. We are grateful for the detailed suggestions that were very helpful for improving it. We hope that you will be satisfied with the revised version in which we have incorporated your points. If you should have any further recommendations for improving our manuscript, please communicate these to us. Please notice that we included additional references into the manuscript. Therefore, the numbers and order of the references in the text as well as in the reference list did changed. These changes are not marked in the manuscript in order to improve readability for the reviewers. For the purpose of your traceability, whenever a reference has been added to the manuscript, it is mentioned in the respective response to your comment. We totally agree with you that the introduction has potential for improvement. Based on our student-focused perspective as researchers involved in the Healthy Campus Mainz project, the introduction of the manuscript is structured as follows to motivate our research questions: i) description of the setting of interest (university students), ii) description of the public health relevance of accidents, iii) description of the lack of knowledge with regard to potential risk groups, iv) description why this is relevant (see your comment number 2), and v) description of the aim of the study. Following yours as well as the first reviewer’s comments on our introduction, we now added the following changes to the introduction: A new paragraph has been added where the scientific relevance of our aim of identifying potential risk-groups for the occurrence of a study-related accident is clarified. Therefore, please see our changes in line 106 - 113 as well as in line 87 - 93. Furthermore, we now give a definition of “study-related accidents” and included corresponding literature (line 66 – 69). In addition, to keep the wording consistent through the whole manuscript, we now consistently use the terms “study-related accident”, “study-related work accident”, and “study-related commuting-accident”. Please see our changes through the whole manuscript. If possible, please also take a look at our responses to the comments two, three, four, and six made by the first reviewer. 

Comment 2: The discussion is also poorly explored with the international literature. This section should be improved in order to explore and discuss the main findings with other works reviewed in section 1 (1. Introduction). 

Response: Following your comment, we now incorporated all citations mentioned in the introduction into the discussion section of the manuscript (references 8-10) and slightly adapted the first paragraph of the discussion with respect to the changes made in the introduction. As you may see, these papers referring to references 8-10 mentioned in the introduction present percentages of study-related accidents based on student surveys (using paper-and-pencil or online questionnaires). For example, 22% of the surveyed sample experienced study-related accident. Since we analyzed objective accident data for the present study (only “real” cases of accidents), our results are poorly comparable with to the results of the survey studies and that is why we did not incorporate these into the discussion section of the manuscript. In the introduction, these references were used to demonstrate the described knowledge gap, the motivation of our study.For the discussion of our findings, we therefore searched for studies focusing on specific student populations (e.g. medical students, students participating to study-related sports activities) and incorporated these into the discussion section of our manuscript. Given that the topic study-related accidents among university students is rarely investigated, we think that we provided a significant literature base within our manuscript. For further clarification, if possible please see also our response to comment number four and our changes in line 87 – 93 of the manuscript referring to this issue. However, in order to address your comment as well as the comment number six made by reviewer one, we now added a new paragraph highlighting the public health relevance of cycling accidents into the introduction section (line 93 – 98). 

Comment 3: Line 307: should be standardized throughout the text. [36–38] should be [36,38]

Response: Thank you for that hint. According to the IJERPHs author instructions (https://www.mdpi.com/journal/ijerph/instructions) it is recommended that, whenever more than two references are mentioned in a square bracket, a hyphen should be used between the lowest and highest reference number. However, with regard to the present comment as well as comment number five, we once again checked the format of the whole manuscript.

Comment 4: The conclusion must be deepened. It is very poor and should be further developed.

Response: We totally agree with you that the conclusion has potential for improvement. Following your comment, we now revised the conclusion.

Comment 5: Authors should follow Instructions for Authors of the journal Int. J. Environ. Res. Public Health.

Response: We totally agree. Please see our response to your comment number three. Once again, we hope that we addressed all comments to your satisfaction and that the revised version of the manuscript is now suitable for publication. In particular, we hope that you accept our argumentation with respect to your comment number two. Should you have any further comments for possible improvements, please communicate these to us.

Round 2

Reviewer 2 Report

Authors should improve tables 2 and 3. Repeat table headings 2 and 3 on subsequent pages. When working with a very long table, it will be executed on several pages. You can configure the table so that the table header row or rows appear automatically on each page.

Author Response

Thank you very much for your comment. We hope that you will be satisfied with the revised version in which we have incorporated the changes. Following your comment, the subheadings (we think that is what was meant by your comment) of table 2 and 3 are now repeated on each page. If you should have any further recommendations for improving our manuscript, please communicate these to us. Kind regards, the authors of the present paper.